# Mapping 1+1-dimensional black hole thermodynamics to finite volume effects

Jean Alexandre[1], Drew Backhouse[1], Eleni-Alexandra Kontou[2], Diego Pardo Santos[2*] and Silvia Pla[1,3]

**1** Department of Physics, King's College London, Strand, London, WC2R 2LS, United Kingdom
**2** Department of Mathematics, King's College London, Strand, London WC2R 2LS, United Kingdom
**3** Physik-Department, Technische Universität München, James-Franck-Str., 85748 Garching, Germany

⋆ diego.pardo@kcl.ac.uk

## Abstract

Both black hole thermodynamics and finite volume effects in quantum field theory violate the null energy condition. Motivated by this, we compare thermodynamic features between two 1+1-dimensional systems: *(i)* a scalar field confined to a periodic spatial interval of length $a$ and tunneling between two degenerate vacua; *(ii)* a dilatonic black hole at temperature $T$ in the presence of matter fields. If we identify $a \propto T^{-1}$, we find similar thermodynamic behaviour, which suggests some deeper connection arising from the presence of non-trivial boundary conditions in both systems.

# 1 Introduction

One of the seminal results in semi-classical gravity is Hawking radiation, and subsequent black hole evaporation [1]. Part of its importance is that it shows a clear quantum effect in a regime where the curvature is small (the black hole horizon) and thus it can be treated classically. An important consequence of Hawking radiation is the violation of the null energy condition (NEC) around the black hole horizon.

The NEC is part of the classical energy conditions which are what we call *pointwise*: they restrict some contraction of the stress tensor at every spacetime point (see [2] and [3] for reviews). The NEC is the weakest of them and it is written as

$$T_{\mu\nu}\ell^{\mu}\ell^{\nu} \geq 0, \tag{1}$$

where $T_{\mu\nu}$ is the stress-energy tensor and $\ell^{\mu}$ a null vector field. The NEC is obeyed by most classical fields[1] and it is often considered an important property of physical matter. Its geometric form, obtained by the use of the Einstein equation, is called the null convergence condition and it implies that a non-rotating null geodesic congruence locally converges. It was famously used in Penrose's singularity theorem [6], Hawking's black hole area theorem [7] and other classical relativity results. If the stress-energy tensor has the form of a perfect fluid, $T_{\mu\nu} = (\rho + p)v^{\mu}v^{\nu} + pg^{\mu\nu}$, where $\rho$ is the energy density, $p$ the pressure and $v^{\mu}$ is the fluid's unit four-velocity vector field, the NEC becomes

$$\rho + p \geq 0. \tag{2}$$

The NEC, as is the case for all pointwise energy conditions, is violated in the context of semi-classical gravity; with the most prominent case being the Hawking radiation. More generally, quantum field theories (QFTs) obeying some reasonable axioms always have states that admit negative energy as shown by Epstein, Glaser and Jaffe in the 1960's [8].

Interestingly, the NEC is violated in a different setting, involving finite volume effects in scalar QFT. A finite volume in QFT implies two fundamental features: quantisation of momentum and tunnelling between multiple vacua. The first feature is at the origin of the Casimir

---

[1]It is however violated by scalars with non-minimal coupling to gravity [4,5].

| System | NEC violation | Entropy rate |
|---|---|---|
| Tunneling in 1+1 flat spacetime | $-\dfrac{\pi}{3a^2}$ | $\dfrac{1}{2} - \dfrac{2m}{3}a$ |
| 1+1 Dilatonic Black Holes | $-\dfrac{N\pi^2}{12}T^2$ | $-\dfrac{N}{12} - \dfrac{M}{\pi T}$ |

Table 1: Summary of results comparing the two thermodynamical systems, for NEC violation and the rate of change in entropy: *(i)* Tunneling in 1+1 flat spacetime (cf. eqs. (31) and (95)); *(ii)* 1+1 Dilatonic Black Holes (cf. eqs. (68) and (96)). $N$ is the number of massless scalar fields.

effect (see [9] for a review), which is known to induce NEC violation. As shown more recently and reviewed in the next section, the second feature also leads to NEC violation, in relation to convexity of the effective potential [10–14]. For both the Casimir effect and tunnelling, NEC violation arises from a ground state energy which is not extensive, i.e. not simply proportional to the size of the system.

We investigate the possibility of a correspondence between these two sources of NEC violation. In particular: *(i)* a scalar field confined to a periodic spatial interval of length $a$ and tunnelling between two degenerate vacua in the limit of zero temperature; *(ii)* a dilatonic black hole at temperature $T$ in the presence of matter fields in an infinite spatial volume.

For simplicity, the particular systems we are considering are both 1+1-dimensional.[2] The motivation is to find common features between two non-trivial thermodynamical systems due to quantum effects. Our main results are summarised in table 1, in the regime $ma \propto M/T \lesssim 1$, where $m, a$ are respectively the mass and length scales in the tunnelling description, and $M, T$ are respectively the mass and temperature of the dilatonic black hole (which are independent parameters). The two systems are thermodynamically similar under the matching condition of $a \propto 1/T$, suggesting a mapping between finite size and finite temperature. It is important to note that this analogy cannot be attributed to dimensional considerations though, since several length/mass scales are present in both models.

We stress here that this work does not establish rigorously a duality between the two systems, as one could hope from the AdS/CFT correspondence for example. The aim of this approach is to put forward a complementary study, which could provide a different angle on black hole thermodynamics, based on an analogy with a simpler system in flat spacetime. Our strategy is to first derive new properties, both for the confined scalar field and the dilatonic black hole, but independently. The resulting mapping $T \leftrightarrow a^{-1}$ we then find is not trivial, and suggests that further studies could be made, requiring a more systematic formalism. Our results are therefore preliminary, but promising for a new and original mapping

In section 2 we calculate the free energy for the ground state resulting from the scalar field tunnelling between degenerate minima. We explain why in one space dimension, the Casimir and tunnelling effects are of the same order of magnitude. In the limit of vanishing temperature, NEC violation can be decomposed as the sum of two contributions: one from the Casimir effect and one from tunnelling. We show that the latter is actually more important than the former if $ma \gtrsim 1$, which is a new feature with relevance potentially going beyond the present study.

Section 3 presents the derivation of the thermodynamical properties of the dilatonic black hole in the presence of non-self-interacting matter fields. A detailed explanation is given for

---

[2]Other studies of finite volume QFT effects in 1+1 dimensional spacetime can be found in [15].

the role of the environment regarding the entropy of the system once backreaction of the
matter fields on the background metric is taken into account.

   In section 4 we compare the two studies and we find that the relation $a \propto T^{-1}$ provides a
mapping between the two systems.

   Tunnelling at finite temperature is described with a Euclidean metric whereas the metric
sign convention for the black hole description is $(-, +)$. Natural units of $\hbar = c = 1$ are used
throughout.

# 2   Finite size effects in 1+1 dimensional flat spacetime

We consider a massive self-interacting real scalar field theory defined on a one-dimensional
periodic interval $x \in [0, a]$ at a temperature $T \equiv 1/\beta$ with a corresponding Euclidean action

$$I = \frac{1}{2} \int_0^\beta \mathrm{d}\tau \int_0^a \mathrm{d}x \left( (\dot{\phi})^2 + (\phi')^2 + \frac{m^2}{4}(\phi^2 - 1)^2 \right) , \tag{3}$$

where $m$ is the mass scale of the theory. Starting from this we study the thermodynamics of
the true vacuum of the effective theory.

## 2.1   Convexity from tunnelling

It is known that the one-Particle-Irreducible (1PI) effective potential is necessarily convex if
one takes several vacua into account [16–25]. Focusing on two degenerate vacua at $\pm v$, the
dynamics of this feature relies on tunnelling between these vacua [10], which restores sym-
metry and induces a true ground state corresponding to a vanishing field expectation value
$\langle \phi \rangle = 0$ (see figure 1). Equivalently, by symmetry of the bare potential, the true vacuum
corresponds to a vanishing source $j = 0$, and

$$\langle \phi \rangle \equiv -\frac{1}{Z[j]} \left. \frac{\delta Z[j]}{\delta j} \right|_{j=0} = 0 , \tag{4}$$

where $Z[j]$ is the partition function. The picture described here can be interpreted as back-
reaction: the double-well bare potential allows quantum fluctuations to tunnel between the
minima, which in turns modifies the vacuum structure by imposing convexity. The resulting
symmetric vacuum corresponds then to a non-perturbative process, which can be described by
the semi-classical approximation for $Z[j]$, as explained below.

   Symmetry restoration is possible in a finite volume only, though, since an infinite volume
implies Spontaneous Symmetry Breaking instead. But a finite volume requires a discrete set
of momenta for quantum fluctuations and, as we show in this article, in the limit of vanishing
temperature, finite-size effects can be decomposed as the sum of two contributions: *(i)* dis-
cretisation of momentum for quantum fluctuations, that we will refer to as the Casimir effect;
*(ii)* symmetry restoration due to tunnelling, that we will refer to as the tunnelling effect.

   Allowing tunnelling between two degenerate vacua, the 1PI effective potential induced by
a dilute gas of instantons was calculated in [11], based on an expansion in $\langle \phi \rangle$ to the quadratic
order. This result explicitly shows a convex effective potential, with a positive mass term and
a true vacuum at $\langle \phi \rangle = 0$. Focusing on this true vacuum, the complete one-loop quantisation
of the dilute gas with discrete momentum is calculated in [14] for a 3-torus, providing the full
picture of the interplay between Casimir and tunnelling effects.

   These calculations were done in 3+1 dimensions though, and we consider here the 1+1
dimensional case, where both effects are comparable. Indeed, for a finite length $a$ and a mass

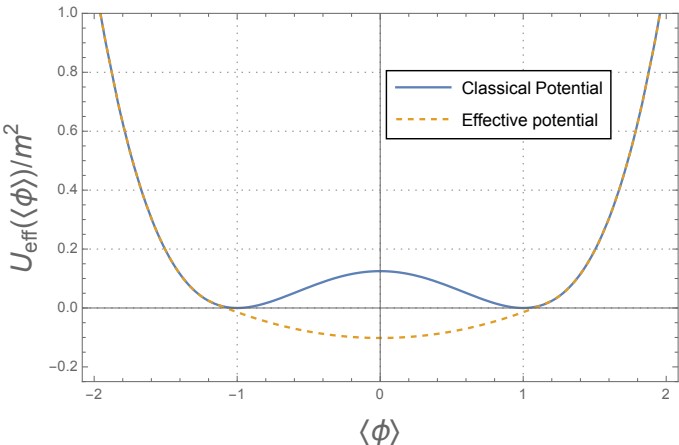

Figure 1: The classical potential (solid blue) and effective potential (dashed orange) in a finite spatial volume. The effective potential is necessarily convex due to tunneling between the two degenerate vacua, restoring symmetry.

scale $m$, the instanton action is of the order $ma$, leading for $ma \gg 1$ to a suppression of the tunnelling effect of the order $\exp(-ma)$, similarly to what happens in the Casimir effect.

As we show in this section, one feature of convexity obtained from quantum fluctuations is NEC violation in the true ground state. If we allowed the system to evolve freely, this violation would imply an increase in the length $a$, similarly to spacetime expansion due to tunnelling-induced NEC violation, as described in [12,13] (see for example [26,27] for reviews of NEC violation in the context of Cosmology). In the present work, we do not take into account spacetime dynamics though, and we stick to static QFT. This implies that some external system fixes the length $a$, which requires some energy. As a consequence, although the confined scalar field violates the NEC, the Averaged NEC is not violated, which can be seen by integrating the NEC along a null geodesic going through the confining walls [28–30].

The main part of this section focuses on the zero-temperature case, since thermal effects tend to restore the NEC. Nevertheless we start the calculations at finite temperature $\beta^{-1}$, and consider then the limit $\beta \to \infty$.

## 2.2 Tunneling and the Casimir effect

### 2.2.1 Semi-classical approximation and true vacuum

Defining the dimensionless variables

$$t \equiv m\tau \quad \text{and} \quad r \equiv mx \,, \tag{5}$$

the action of eq. (3) becomes

$$I = \frac{1}{2} \int_0^{m\beta} \mathrm{d}t \int_0^{ma} \mathrm{d}r \left( (\dot{\phi})^2 + (\phi')^2 + \frac{1}{4}(\phi^2 - 1)^2 \right) \,, \tag{6}$$

where dots and primes now represent derivatives in $t$ and $r$ respectively. One can see that this action depends on the two dimensionless parameters $ma$ and $m\beta$, and is invariant under the simultaneous rescaling

$$a \to \lambda a \,, \quad \beta \to \lambda \beta \,, \quad m \to m/\lambda \,, \tag{7}$$

and the quantum theory should also respect this symmetry, as we confirm in what follows. The equation of motion (EoM) with solutions $\phi_i$ is

$$\ddot{\phi}_i + \frac{1}{2} \left( \phi_i - \phi_i^3 \right) = 0 \,, \tag{8}$$

150   where in this work we consider only Euclidean-time-dependent and homogeneous configu-
151   rations $\phi_i$, since vacuum bubbles forming from degenerate vacua would have an infinite ra-
152   dius [31, 32]. There are several solutions to the classical EoM (8), each to be studied in
153   following subsections.

154      For a vanishing source $j = 0$ and in the semi-classical approximation, the partition function
155   $Z[0] \equiv Z$ can be approximated as a sum of path integrals over regions in field space around
156   the action minimising saddle points $\varphi_i$ via the field decomposition $\varphi = \varphi_i + \psi$, integrating
157   over fluctuations $\psi$. This assumes that the fluctuations do not overlap, and the one-loop
158   approximation for the fluctuation factors leads to

$$
\begin{aligned}
Z &= \int \mathcal{D}[\phi] \, \exp\left(-I[\phi]\right) & (9) \\
&\simeq \sum_i \int \mathcal{D}[\psi] \, \exp\left(-I[\varphi_i + \psi]\right) \\
&= \sum_i \left(\det(\delta^2 I[\phi_i])\right)^{-1/2} \exp\left(-I[\phi_i]\right) \\
&\equiv \sum_i \exp\left(-W[\phi_i]\right),
\end{aligned}
$$

159   where the individual connected graph generating functionals are

$$
W[\phi_i] \equiv I[\phi_i] + \frac{1}{2}\,\mathrm{Tr}\left\{\ln\left(\delta^2 I[\phi_i]\right)\right\} . \tag{10}
$$

### 2.2.2   Static saddle points

161   There are two static saddle points in the present model, $\phi_s = \pm 1$. The corresponding indi-
162   vidual connected graph generating functionals (10) can be evaluated using known methods
163   developed for the study of the thermal Casimir effect on a 1D periodic interval [9]. The steps
164   are outlined in appendix A and lead to

$$
W_{\mathrm{stat}}(a,\beta) \equiv W[\phi_s] = a\beta\Lambda^2 - \frac{m^2 a\beta}{\pi}\int_1^\infty \mathrm{d}u\,\frac{\sqrt{u^2-1}}{e^{mau}-1} + \sum_{n\in\mathbb{Z}}\ln\left(1-e^{-\beta\omega_n}\right) , \tag{11}
$$

165   where $\Lambda^2$ is an ultraviolet cutoff, corresponding to the vacuum energy of unbounded space,
166   and the quantised frequencies/wave vectors are

$$
\omega_n = \sqrt{m^2 + k_n^2} \quad , \quad k_n = \frac{2\pi n}{a} . \tag{12}
$$

167   We note here that quantum corrections indeed depend on $a, \beta, m$ through the products $ma$
168   and $m\beta$ only.

### 2.2.3   Time-dependent saddle points

170   The fundamental time-dependent saddle point is the (anti-)instanton relating the two vacua
171   of the bare potential

$$
\phi_{\mathrm{inst}}(t) = (\pm)\tanh\left(\frac{t - t_1}{2}\right) , \tag{13}
$$

172   where $t_1$ is the time of the jump. At a finite temperature, field configurations are periodic in
173   Euclidean time and hence instantons and anti-instantons can only exist in pairs. Such field

configurations are well approximated by a product of individual (anti-)instanton configurations

$$\phi_{\text{n-pair}}(\tau) \simeq \prod_{j=1}^{2n} (-1)^j \tanh\left(\frac{t-t_j}{2}\right) , \tag{14}$$

provided that the jumps at $t_i$ and $t_j$ are sufficiently distant ($|t_i - t_j| \gg 1$). The factor $-1$ ensures that an instanton is always followed by an anti-instanton and the product is taken to $2n$ to enforce periodicity in Euclidean time. In the limit of small temperature, $m\beta \gg 1$, a large amount of instanton/anti-instanton pairs is allowed and we assume in what follows the instanton dilute gas approximation [33], where the width of each jump is negligible compared to $m\beta$. Also, (anti-)instantons are far enough from each other for them to keep their shape, which for $n$ pairs leads to the total action

$$I_{\text{n-pairs}} \simeq 2n I_{\text{inst}} , \tag{15}$$

where the action for one (anti-)instanton is

$$I_{\text{inst}} \equiv I[\phi_{\text{inst}}] = \frac{2ma}{3} . \tag{16}$$

The fluctuation factor for $n$ instanton/anti-instanton pairs can then be approximated by the product of fluctuation factors for each static saddle point evaluated over half the total Euclidean time interval $\beta/2$, times the fluctuation factors for each instanton jump. The corresponding connected graph generating functional is then

$$W_{\text{n-pairs}} \equiv W[\phi_{\text{n-pair}}] \simeq 2W_{\text{stat}}(a, \beta/2) + 2n W_{\text{jump}} , \tag{17}$$

where we know from tunnelling in Quantum Mechanics [33] that

$$W_{\text{jump}} \equiv I_{\text{inst}} - \frac{1}{2}\ln\left(\frac{6I_{\text{inst}}}{\pi}\right) . \tag{18}$$

We note that the expression for $W_{\text{jump}}$ takes into account time-dependent quantum fluctuations over the instantons, and it neglects the space-dependence of these fluctuations. However, it was shown in [14] that the main contribution of the instanton jump comes from the zero-modes, validating the approximation made here.

### 2.2.4 Partition function

Assuming the semi-classical approximation and a dilute gas of instantons/anti-instantons, the partition function can be expressed as a sum over all the possible $n$-pair configurations

$$Z \simeq 2\exp\left(-W_{\text{stat}}(a, \beta)\right) + 2\sum_{n=1}^{\infty}\left(\prod_{i=1}^{2n}\int_{t_{i-1}}^{m\beta} dt_i\right)\exp\left(-2W_{\text{stat}}(a, \beta/2) - 2n W_{\text{jump}}\right) . \tag{19}$$

The first term in the right-hand side corresponds to the two static saddle points. In the second term the product of integrals accounts for the invariance of the total action $I_{\text{n-pairs}}$ under the translations of each successive instanton jump over the remaining dimensionless Euclidean time interval $t \in [t_i, m\beta]$, defining $t_0 \equiv 0$ since the first instanton can exist over the whole interval. This invariance under translation of the jumps corresponds to the zero modes of the fluctuation factors for each $n$−pair configuration. The factor 2 takes into account the instanton configurations starting and ending at either $+1$ or $-1$. Using the known result [31]

$$\prod_{i=1}^{2n}\int_{t_{i-1}}^{m\beta} dt_i = \frac{(m\beta)^{2n}}{(2n)!} , \tag{20}$$

the partition function (19) can be expressed as

$$Z \simeq 2 \exp\left(-W_{\text{stat}}(a, \beta)\right) + 2 \exp\left(-2 W_{\text{stat}}(a, \beta/2)\right) \sum_{n=1}^{\infty} \frac{\bar{N}^{2n}}{(2n)!} \,, \tag{21}$$

where

$$\bar{N} \equiv m\beta \sqrt{\frac{6 I_{\text{inst}}}{\pi}} e^{-I_{\text{inst}}} = 2 m\beta \sqrt{\frac{ma}{\pi}} \, \exp\left(-\frac{2ma}{3}\right) \,, \tag{22}$$

with $\bar{N}/2$ corresponding to the average number of instanton/anti-instanton pairs over the whole Euclidean time $\beta$ [10]. In the small temperature limit ($m\beta \gg 1$) the last term in $W_{\text{stat}}$ (11) can be neglected, such that it can be taken as linear in $\beta$

$$W_{\text{stat}}(a, \beta) \simeq 2 W_{\text{stat}}(a, \beta/2) \,, \tag{23}$$

and the partition function (21) becomes [3]

$$Z \simeq 2 \exp\left(-2 W_{\text{stat}}(a, \beta/2)\right) \sum_{n=0}^{\infty} \frac{\bar{N}^{2n}}{(2n)!} = 2 \exp\left(-2 W_{\text{stat}}(a, \beta/2)\right) \cosh(\bar{N}) \,. \tag{24}$$

Finally, in the limit $m\beta \gg 1$ for finite $ma$ (such that $\bar{N} \gg 1$) the total free energy is

$$F_{\text{true}} \equiv -T \ln(Z) \simeq 2 T W_{\text{stat}}(a, \beta/2) - T\bar{N} \,, \tag{25}$$

and corresponds to the sum of the usual free-field Casimir contribution $2 T W_{\text{stat}}(a, \beta/2)$ and the tunneling contribution $-T\bar{N}$.

## 2.3 Null Energy Condition

We show here that the true ground state of the system we consider violates the NEC, as a consequence of the true vacuum energy not being extensive: the free energy (25) is not simply proportional to $a$.

   We assume here that the dilute instanton gas described by the partition function (24) may be treated as a perfect fluid, such that the resulting null energy condition reduces to the simpler form (2). The thermodynamic energy density and pressure are then defined as

$$\rho \;\equiv\; \frac{1}{a}\left(F_{\text{true}} + \beta \frac{\partial F_{\text{true}}}{\partial \beta}\right), \tag{26}$$

$$p \;\equiv\; -\frac{\partial F_{\text{true}}}{\partial a} \,. \tag{27}$$

Here we show that the NEC is violated by the finite volume effects we consider.

   The sum $\rho + p$ may be evaluated from the free energy (25) and satisfies

$$\begin{aligned}
\frac{\rho + p}{m^2} \;=\; & -\frac{ma}{\pi} \int_1^{\infty} du \frac{u e^{mau} \sqrt{u^2 - 1}}{(e^{mau} - 1)^2} - \frac{4ma + 3}{3\sqrt{\pi ma}} \exp\left(-\frac{2ma}{3}\right) \\
& + \sum_{n \in \mathbb{Z}} \frac{m^2 a^2 + 8 n^2 \pi^2}{m^2 a^3 \omega_n} \left(e^{\beta \omega_n/2} - 1\right)^{-1} \,,
\end{aligned} \tag{28}$$

where we can identify the following terms:

---

[3]The approximation (23) is applied to the static saddle point contribution, and not to the instanton contribution, since it is sub-dominant at low temperatures.

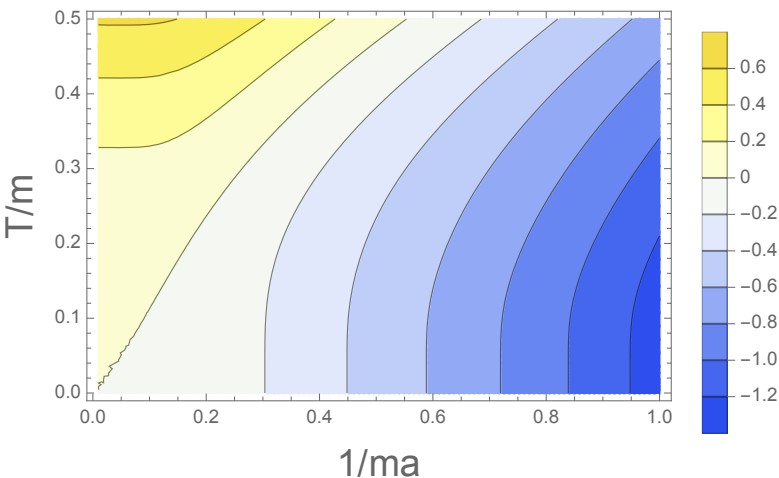

Figure 2: Numerical plot of $(\rho + p)/m^2$ (28) for inverse dimensionless length $1/ma$ with finite temperature corrections. At zero temperature, NEC violation increases as the spatial interval reduces, corresponding to an increased tunneling rate, and approaches zero in the limit of infinite spatial length where tunneling is completely suppressed. For a given length scale, finite temperature effects increase $\rho + p$ and can lead to NEC satisfaction at large length scales.

(i) the first term (integral over $t$) corresponds to the known Casimir effect, obtained for a free field, with a negative contribution;

(ii) the second term corresponds to tunnelling arising from degenerate vacua, with a negative contribution;

(iii) the third term (sum over Matsubara modes) corresponds to finite temperature effects providing a positive contribution, and becomes the usual contribution from black body radiation ($\propto T^2$) in the massless and infinite length limit.

The expression (28) is plotted in figure 2 as a function of inverse dimensionless spatial length $1/ma$ with finite temperature corrections.

Thermal effects decrease exponentially for small temperatures: for $m\beta \gg 1$ we have

$$\sum_{n\in\mathbb{Z}} \frac{m^2 a^2 + 8n^2\pi^2}{m^2 a^3 \omega_n} \left(e^{\beta \omega_n/2} - 1\right)^{-1} \simeq \frac{e^{-m\beta/2}}{am} \, , \tag{29}$$

and in what follows we take the limit $\beta \to \infty$, in order to focus on NEC violating finite-length effects. It is then interesting to look at two asymptotic regimes for the length $a$:

   – $\underline{ma \gg 1}$ In this case we have

$$\frac{\rho + p}{m^2} \approx -\frac{e^{-am}}{\sqrt{2\pi am}} - \frac{4}{3}\sqrt{\frac{am}{\pi}} \, e^{-2ma/3} \, , \tag{30}$$

   and we can see that tunnelling effects are more important than the free-field Casimir effect;

   – $\underline{ma \lesssim 1}$ In this case we have

$$\frac{\rho + p}{m^2} \approx -\frac{\pi}{3(am)^2} \, , \tag{31}$$

   where tunnelling is negligible and the result is identical to the one obtained for a massless free field [9].

 ## 2.4 Entropy

241 At zero temperature the classical thermal entropy vanishes, which can be seen with

$$\mathcal{S}_{\text{classical}} = -\lim_{T \to 0} \frac{\partial F_{\text{true}}}{\partial T} = 0 \ . \tag{32}$$

242 There is a quantum contribution left though, which can be interpreted as the entropy for
243 the dilute gas of instantons/anti-instantons which relate the two vacua. Taking into account
244 the instanton fluctuation factors described above, the probability of having $n$ instanton/anti-
245 instanton pairs may be read off from the partition function $Z$

$$p_n = \frac{2}{Z} \frac{(\bar{N})^{2n}}{(2n)!} \mathrm{e}^{-2W_{\text{stat}}(\beta/2)} = \frac{1}{\cosh(\bar{N})} \frac{(\bar{N})^{2n}}{(2n)!} \ , \tag{33}$$

246 where $Z$, $\bar{N}$ and $W_{\text{stat}}$ are given by eqs. (24), (22) and (11) respectively. The entropy of
247 the dilute gas should be extensive and thus proportional to its number of degrees of freedom
248 (although it is not proportional to the length $a$). The entropy for the systems of instantons
249 and anti-instantons $\mathcal{S}_{\text{tun}}$ is then twice the entropy for the system of pairs, which is given by
250 the usual sum over probabilities

$$\mathcal{S}_{\text{tun}} = -2 \sum_{n=0}^{\infty} p_n \ln(p_n) \ . \tag{34}$$

251 In the limit $\bar{N} \gg 1$, we find numerically that the entropy (34) is asymptotically equivalent to

$$\begin{aligned} \mathcal{S}_{\text{tun}} &\approx \ln(\bar{N}) \\ &= \ln(m\beta) - I_{\text{inst}} + \frac{1}{2} \ln\left(\frac{6}{\pi} I_{\text{inst}}\right) \ , \end{aligned} \tag{35}$$

252 where $I_{\text{inst}}$ is the instanton action as given in (16), and the result matches the usual micro-
253 canonical entropy for $\bar{N}$ microstates. As expected, one can also check that $\mathcal{S}_{\text{tun}}$ vanishes in the
254 limit $a \to \infty$ (where $\bar{N} \to 0$, even for zero temperature), since tunnelling is then completely
255 suppressed. This behaviour is in correspondence with an "effective third law of thermodynam-
256 ics", where $1/a$ plays the role of a temperature. We will come back to this analogy later in this
257 article.

258     For a finite length $a$, the entropy is non-zero with a logarithmic divergence in the zero
259 temperature limit. Such logarithmic divergences in the zero temperature entropy of a quantum
260 system are not new, such as the massless Casimir effect [34] and it was argued in [35] that
261 such divergences should be removed.

262     The isothermal compressibility, defined as

$$K \equiv -\frac{1}{a} \frac{\partial a}{\partial p} \equiv \frac{1}{a} \left( \frac{\partial^2 F_{\text{true}}}{\partial a^2} \right)^{-1} \ , \tag{36}$$

263 is negative for all $a$, which is usually interpreted as a sign of instability. One may think that
264 this instability is similar to the one obtained for a Van der Waals fluid experiencing an isother-
265 mal compression. If one assumes homogeneity of the Van der Waals fluid in a volume $V$, the
266 bulk modulus $-V\partial p/\partial V$ is negative in a given range of volumes, which is not physical. What
267 happens is that the fluid separates into two phases, liquid and vapour, leading to the Maxwell
268 construction, which corresponds to a constant-pressure plateau. The position of this plateau
269 is determined by identifying the chemical potentials in each phase. Also, a constant pressure
270 leads to a vanishing compressibility, and the true free energy is convex, as expected for the

Legendre transform of the internal energy. This constant-pressure plateau allows random re-
gions of one vacuum or the other, in a proportion given by the volume, which varies between
values corresponding to the first drop of liquid and the last bubble of vapour.

In our case, the system remains homogeneous though: the effective potential does not fea-
ture any plateau, but it has a unique minimum at $\langle\phi\rangle = 0$. An intuitive description is provided
by weakly-interacting spins on a lattice, each with a random direction and a vanishing average
value.[4] A flat effective potential would be obtained in the limit $ma \to \infty$, where the tunnelling
rate exponentially vanishes though, in which case one would have to wait an infinite amount
of time for the true vacuum to settle. As mentioned in section 2.1, in our situation the insta-
bility indicated by the negative compressibility would correspond to a spacetime expansion, if
no environment was present to fix the length $a$.

# 3 Black holes in 1+1 dilaton gravity

The study of Hawking radiation including its backreaction on the spacetime geometry is an
extremely difficult problem in $3+1$ dimensions. Motivated by dimensional reduction, it is pos-
sible to simplify the problem by studying the $1+1$ dimensional case. In particular, we focus
on the classical Callan, Giddings, Harvey and Strominger (CGHS) two-dimensional dilatonic
black hole model [36, 37]. For the semi-classical description of the theory including backre-
action, we consider the standard Polyakov term which represents the leading order quantum
fluctuations in a $1/N$ expansion where $N$ is the number of matter fields. To find analytical
solutions to the semiclassical theory it is necessary to introduce suitable counterterms in the
action. To consider these counterterms, we introduce the one-parameter family of models that
ranges between the Russo, Susskind and Thorlacius (RST) model [38] and the Bose, Parker,
Peleg (BPP) model [39], following the parameterization of the action presented in [40]. We
focus our study on the BPP model since it results in simpler expressions for the metric and the
dilaton.

After describing the solutions of the semi-classical theory, this section delves into the im-
plications for the stress-energy tensor and the entropy of two-dimensional black holes.

## 3.1 Introduction to dilaton gravity

In two dimensions, the Einstein-Hilbert action is just the Euler characteristic of the manifold
(accordingly, $G_{\mu\nu}$ vanishes identically), and $1+1$ dimensional gravity is trivial. Since we want
to capture aspects of the $3+1$ dimensional theory within a $1+1$ dimensional description, we
use the dilaton field [41, 42] which emerges from the compactification of higher dimensions.
Here, we derive the dilaton from the dimensional reduction of spherically symmetric gravity
in $3+1$ dimensions.

We consider the 3+1 Einstein-Hilbert action

$$I_{EH}^{(4)} = \frac{1}{16\pi G^{(4)}} \int \mathrm{d}^4 x \sqrt{-g^{(4)}} R^{(4)}, \tag{37}$$

where $^{(4)}$ indicates the spacetime dimension and $G^{(4)}$ is Newton's constant. We consider the
spherically symmetric ansatz

$$\mathrm{d}s_{(4)}^2 = g_{ab}\mathrm{d}x^a\mathrm{d}x^b + \frac{e^{-2\phi(x^a)}}{\lambda^2}(\mathrm{d}\theta^2 + \sin^2\theta\,\mathrm{d}\varphi^2), \tag{38}$$

---

[4]This is different from the high-temperature limit, where thermal fluctuations dominate over spin interactions
and lead to a random spin distribution. The vanishing average spin discussed here happens at zero temperature,
and is due to tunnelling instead

308  where the radius $r$ of the 2-sphere has been parametrized via a dilaton field $\phi(x^a)$, $r = \lambda^{-1}e^{-\phi}$.
309  The parameter $\lambda$ is dimensionful and is introduced to get a dimensionless dilaton. Using this
310  ansatz, we can write $R^{(4)}$ in terms of $R \equiv R^{(2)}$ and the four dimensional volume element in in
311  terms of the two-dimensional volume element times the angular terms [43,44]

$$
\begin{aligned}
R^{(4)} &= R + 2(\nabla\phi)^2 + 2\lambda^2 e^{2\phi} - 2e^{2\phi}\Box e^{-2\phi}, \\
\mathrm{d}^4x\sqrt{-g^{(4)}} &= \mathrm{d}^2x\,\mathrm{d}\theta\,\mathrm{d}\varphi\sqrt{-g}\frac{e^{-2\phi}}{\lambda^2}\sin^2\theta.
\end{aligned}
\tag{39}
$$

312  Integrating out the angular part, the resulting dilaton action is [43,44]

$$
I_D = \frac{1}{4\pi\lambda^2 G^{(4)}}\int \mathrm{d}^2x\sqrt{-g}\left(e^{-2\phi}\left(R + 2(\nabla\phi)^2\right) + 2\lambda^2\right),
\tag{40}
$$

313  where we see that $\lambda$ plays the role of a cosmological constant in the reduced theory. We define
314  a two dimensional Newton's constant $G^{(2)} = \lambda^2 G^{(4)}$ and work in units where $G^{(4)} = \frac{1}{2\lambda^2}$.
315      To simplify the theory in such a way that it is possible to find an exact analytical solution,
316  we work with the action[5]

$$
I_\phi = \frac{1}{2\pi}\int \mathrm{d}^2x\sqrt{-g}e^{-2\phi}\left(R + 4(\nabla\phi)^2 + 4\lambda^2\right),
\tag{41}
$$

317  where we have modified the potential term of the dilation and the coefficient of the kinetic
318  term as compared with (40). Despite the changes, this dilaton theory still has black holes and
319  Hawking radiation [36,37,46]. We work in conformal gauge,

$$
\mathrm{d}s^2 = -e^{2\eta}\mathrm{d}x^+\mathrm{d}x^-,
\tag{42}
$$

320  with null coordinates $x^\pm$. The EoM resulting from the variation of the action with respect to
321  $\eta$ and $\phi$ can be conveniently written in terms of $2(\eta-\phi)$ and $e^{-2\phi}$, namely

$$
\partial_+\partial_- e^{-2\phi} + \lambda^2 e^{2(\eta-\phi)} = 0,
\tag{43}
$$

$$
2e^{-2\phi}\partial_+\partial_-(\eta-\phi) + \partial_+\partial_- e^{-2\phi} + \lambda^2 e^{2(\eta-\phi)} = 0.
\tag{44}
$$

322  Additionally, we derive the following constraints from the variation of the action with respect
323  to the $(\pm, \pm)$ components of the metric $g_{\mu\nu}$

$$
\partial_\pm^2 e^{-2\phi} - 2\partial_\pm(\eta-\phi)\partial_\pm e^{-2\phi} = 0.
\tag{45}
$$

324  Combining equations (44) and (43) we get the free field equation

$$
2\partial_+\partial_-(\eta-\phi) = 0,
\tag{46}
$$

325  which has solutions of the form $2(\eta-\phi) = h(x^+) + s(x^-)$. In the Kruskal gauge, the remaining
326  freedom is fixed by making $h(x^+) = s(x^-) = 0$, i.e. $\eta = \phi$. This model has black hole solutions
327  which, in the Kruskal gauge, are (see [36,43] for a detailed discussion)

$$
\mathrm{d}s^2 = -\frac{dx^+dx^-}{(M/\lambda - \lambda^2 x^+x^-)},
\tag{47}
$$

$$
\eta = \phi = -\frac{1}{2}\ln\left(\frac{M}{\lambda} - \lambda^2 x^+x^-\right),
\tag{48}
$$

---

[5]$I_\phi$ can be exactly derived by integrating out the angular part of near-extreme, magnetically charged black holes
in four-dimensional dilaton gravity [36,45].

328 where $M$ is an integration constant that corresponds to the ADM mass of the black hole [36].
329 This metric has a curvature singularity at $\lambda^2 x^+ x^- = M/\lambda$ and horizons at $\lambda^2 x^+ x^- = 0$, see
330 figure 3. The surface gravity can be easily computed, and reads $\kappa = \lambda$. Therefore, the black
331 hole temperature is

$$T = \frac{\lambda}{2\pi}. \tag{49}$$

332 In contrast to the black hole temperature in four dimensions, the two-dimensional black hole
333 temperature is independent of the mass $M$. We use these results to evaluate the NEC and the
334 entropy.

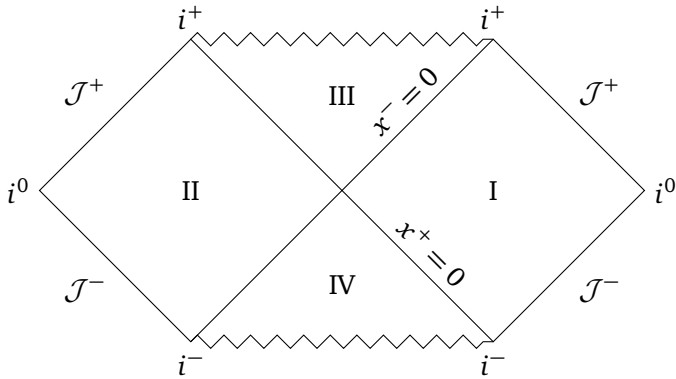

Figure 3: Penrose diagram for a static two-dimensional dilatonic black hole.

### 335 3.2 Adding quantum matter: static black holes

336 Now we add $N$ massless scalar fields $f_i$ and we have the total action

$$I_0 = I_\phi + I_f = \frac{1}{2\pi} \int d^2x \sqrt{-g} \left[ e^{-2\phi} \left( R + 4(\nabla\phi)^2 + 4\lambda^2 \right) - \frac{1}{2} \sum_{i=1}^{N} (\nabla f_i)^2 \right]. \tag{50}$$

337 This action corresponds to the CGHS model. We continue the analysis in conformal gauge
338 (42). In these coordinates, the classical stress-energy tensor for the fields $f_i$ is

$$T_{\pm\pm} = \frac{1}{2} \sum_{i=0}^{N} (\partial_\pm f_i)^2. \tag{51}$$

339 The next step is to quantise the theory. We want to focus on static solutions, $\langle f_i \rangle = 0$, and look
340 at the one-loop quantum corrections to the stress-energy tensor in different vacuum states. The
341 quantum corrections of the different fields that we have seen in the CGHS model contribute to
342 the semiclassical theory. In order to make the analysis feasible, we assume that the number of
343 matter fields $N$ is very large and calculate the effective action at leading order in an expansion
344 in $1/N$. In this limit, the quantum fluctuations of the dilaton and the metric can be ignored
345 and we only have to consider the one-loop correction of the matter fields to the stress-energy
346 tensor [37, 47, 48]. This one-loop correction to $T_{\mu\nu}$ due to the $N$ massless scalar fields can
347 be evaluated using the trace anomaly, which relates the expectation value of the stress-energy
348 tensor and the Ricci scalar [49]

$$\langle T \rangle = \frac{N}{24} R. \tag{52}$$

349 In conformal gauge, the trace anomaly leads to

$$\langle T_{+-} \rangle = -\frac{N}{12} \partial_+ \partial_- \eta. \tag{53}$$

350 In addition, we can use $\langle T_{+-}\rangle$ in (53) together with the conservation of the stress-energy tensor
351 to determine $\langle T_{\pm\pm}\rangle$

$$\langle T_{\pm\pm}\rangle = -\frac{N}{12}\left(\partial_\pm\eta\partial_\pm\eta - \partial_\pm^2\eta + t_\pm\right), \tag{54}$$

352 where $t_\pm$ is fixed by boundary conditions (vacuum choice). We will analyse two vacuum
353 choices: the Hartle-Hawking vacuum, which describes thermal equilibrium at infinity and is
354 given by $t_\pm = 0$, and the Boulware vacuum, which describes empty space at infinity and is
355 given by the boundary conditions $t_\pm = -\frac{1}{4(x^\pm)^2}$.
356    Alternatively, the expectation value of the stress-energy tensor can be obtained by func-
357 tional differentiation of an effective action, the Polyakov action

$$\langle T_{\mu\nu}\rangle = -\frac{2\pi}{\sqrt{-g}}\frac{\delta I_P}{\delta g^{\mu\nu}}, \tag{55}$$

358 where

$$I_P = -\frac{N}{96\pi}\int d^2x\sqrt{-g(x)}\int d^2y\sqrt{-g(y)}R(x)G(x,y)R(y). \tag{56}$$

359 $G(x,y)$ is the Green's function for the differential operator $\Box_g$. $I_P$ incorporates the backre-
360 action of the quantum fluctuations of the matter fields on the metric. By writing $I_P$ in the
361 conformal gauge, we can derive immediately Eqs. (53) and (54) from (55). We will use these
362 expressions later on in the evaluation of the NEC.
363    It is convenient to convert the non-local Polyakov action into a local one by introducing an
364 auxiliary scalar field $\varphi$ constrained to obey the equation $\Box_g\varphi = R$ (see for example eq. (5.56)
365 in [43]). By doing this, the local action is

$$I_P = -\frac{N}{96\pi}\int d^2x\sqrt{-g(x)}\left(\varphi\Box_g\varphi + 2\varphi R\right). \tag{57}$$

366 In conformal gauge, the equation of motion for $\varphi$ has the following solution

$$\varphi(x^\pm) = -2\eta(x^\pm) + 2\left(\varphi_+(x^+) + \varphi_-(x^-)\right), \tag{58}$$

367 where $\varphi_+(x^+)$ and $\varphi_-(x^-)$ are solutions of

$$-(\partial_\pm\varphi_\pm)^2 + \partial_\pm^2\varphi_\pm = t_\pm(x^\pm). \tag{59}$$

368    We are also interested in the entropy of the system. As we showed in (47), this theory
369 has black hole solutions. Therefore, the total entropy of the system consists of two terms: the
370 geometric black hole entropy, which is the $1+1$ dimensional equivalent of the Bekenstein-
371 Hawking entropy, and the von Neuman entropy, associated to the quantum fields outside the
372 horizon, usually called fine-grained entropy [37, 50, 51]. For general diffeomorphism invari-
373 ant theories, it is possible to evaluate both of these entropies using the method developed in
374 ref. [52] and particularised to the context of two-dimensional gravity in Refs. [50, 51, 53]. In
375 this way, both entropies can be evaluated in a geometrical way by calculating the derivatives
376 of the Lagrangian associated with the different contributions to the action with respect to the
377 curvature

$$\mathcal{S}_\phi = \left.\frac{4\pi}{\sqrt{-g}}\frac{\partial\mathcal{L}_\phi}{\partial R}\right|_H = \left.2e^{-2\phi}\right|_H \tag{60}$$

$$\mathcal{S}_P = \left.\frac{4\pi}{\sqrt{-g}}\frac{\partial\mathcal{L}_P}{\partial R}\right|_H = \left.-\frac{N}{12}\varphi\right|_H, \tag{61}$$

378 where $|_H$ means that these quantities should be evaluated at the horizon. It can be shown
379 that $\mathcal{S}_\phi$ and $\mathcal{S}_P$ are equivalent to the black hole and the fine-grained entropy respectively
380 [37, 51, 54]. Equivalently, the entropy can be evaluated using the Euclidean path integral
381 approach, as done in ref. [55] for the RST model.

382 In what follows, we proceed to evaluate the NEC and the entropy. We start with the simpler
383 case without backreaction, and then we study how backreaction modifies the results.

## 3.3 Without backreaction, the CGHS model

385 For the case without backreaction, the background spacetime is described by (50), i.e. the
386 CGHS model. Since we are interested in static black holes, we focus on the case where $\langle f \rangle = 0$.
387 As we have seen, under these conditions, the solution is an eternal black hole of mass $M$.

### 3.3.1 Null energy condition

389 The null energy condition

$$\langle T_{\mu\nu} \rangle \ell^\mu \ell^\nu \geq 0 \,, \tag{62}$$

390 gives two equations that, in the conformal gauge, are proportional to the diagonal components
391 of the stress-energy tensor $\langle T_{\pm\pm} \rangle$ . In what follows, we will focus on the exterior region, so
392 we assume $x^+ > 0$ and $x^- < 0$. For the Hartle-Hawking vacuum ($t_\pm = 0$), using the dilatonic
393 metric (48) in eq. (54), we find[6]

$$NEC_H^\pm = (\lambda x^\pm)^2 \langle H|T_{\pm\pm}|H \rangle = \frac{N\lambda^2}{48} (\lambda x^\pm)^2 (\lambda x^\mp)^2 e^{4\eta} > 0 \,, \tag{63}$$

394 while for the Boulware vacuum we obtain

$$NEC_B^\pm = (\lambda x^\pm)^2 \langle B|T_{\pm\pm}|B \rangle = -\frac{N\lambda^2}{48} \left(1 - (\lambda x^\pm)^2 (\lambda x^\mp)^2 e^{4\eta}\right) < 0 \,. \tag{64}$$

395 We can easily check that the difference between the Hartle–Hawking and Boulware stress
396 energy tensors is just a thermal distribution of massless particles at the Hawking temperature[7]

$$NEC_H^\pm - NEC_B^\pm = N\frac{\lambda^2}{48} = N\frac{\pi^2}{12} T^2 \,, \tag{65}$$

397 where $T = \lambda/2\pi$ is the black hole temperature as given in eq. (49). The extra $N$ factor
398 appears because we are considering $N$ conformal fields. For convenience, we can write the
399 stress-energy tensor in the Hartle-Hawking vacuum as

$$NEC_H^\pm = \frac{N\lambda^2}{48} - \frac{N\lambda^2}{48} \left(1 - (\lambda x^\pm)^2 (\lambda x^\mp)^2 e^{4\eta}\right) > 0 \,. \tag{66}$$

400 In this expression, we see that there is a negative contribution coming from pure vacuum
401 effects plus a positive contribution coming from thermal effects. As a final note, we point out
402 that the quantity $\left(1 - (\lambda x^\pm)^2 (\lambda x^\mp)^2 e^{4\eta}\right)$ vanishes as $x^\pm \to \pm\infty$ and tends to 1 as $x^\pm \to 0$.
403 It means that $(\lambda x^+)^2 \langle H|T_{++}|H \rangle \to N\lambda^2/48$ asymptotically (constant thermal flux), and goes
404 to zero at the horizon (thermal bath in thermal equilibrium with the black hole, so the fluxes

---

[6]The condition $\langle T_{\mu\nu} \rangle \ell^\mu \ell^\nu \geq 0$ is defined up to a positive overall factor. In this case, we find it useful to compute $(\lambda x^\pm)^2 \langle T_{\pm\pm} \rangle$ instead of $\langle T_{\pm\pm} \rangle$, since it is the quantity related to an asymptotic observer at infinity. This quantity comes directly from the transformation law for tensors $T_{\nu\nu} = (dx^+/d\nu)^2 T_{++}$ (and similarly for the other components).

[7]See ref. [56] for a similar analysis in a Schwarzschild background.

cancel). We can remove the contribution from the thermal bath to compute the NEC vacuum contribution of the black hole

$$NEC_{\text{BH}} = (\lambda x^{\pm})^2 \langle H|T_{\pm\pm}|H\rangle - \left(\frac{N\lambda^2}{48}\right) = -\frac{N\lambda^2}{48}\left(1 - (\lambda x^{\pm})^2(\lambda x^{\mp})^2 e^{4\eta}\right) < 0, \qquad (67)$$

which results in a negative contribution to the NEC due to vacuum effects. On the horizon

$$NEC_{\text{BH}} = -\left(\frac{N\pi^2 T^2}{12}\right). \qquad (68)$$

In figure 4 we summarise the results for the NEC.

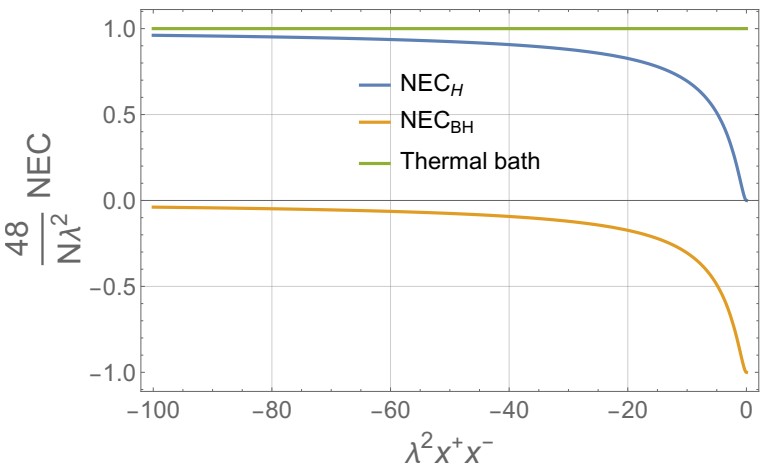

Figure 4: We plot the (normalized) energy condition $\frac{48}{N\lambda^2}$ (NEC) as a function of $\lambda^2 x^+ x^-$ for the BPP and classical models. The green curve corresponds to the thermal bath eq. (65), the orange curve corresponds to the contribution from vacuum effects eq. (64), and the total NEC eq. (66) is represented in blue. The black hole mass is $M/\lambda = 2$.

### 3.3.2 Entropy and thermodynamics

Inserting the dilaton solution (48) into (60), the black hole entropy is

$$\mathcal{S}_{BH} = \mathcal{S}_{\phi} = \frac{2M}{\lambda} = \frac{M}{\pi T}. \qquad (69)$$

From the dimensional reduction (38) relating the radius and the dilaton, we can interpret this term as $1/4$ of the horizon area of the classical $3+1$-dimensional black hole. Remember that in the dimensional reduction (38), $r = \lambda^{-1}e^{-\phi}$ and $\mathcal{S}_{\phi} = 2e^{-2\phi_H} = \frac{2\lambda^2}{\pi}\frac{4\pi r_H^2}{4}$ (see eq. (60)).

To evaluate the entropy associated with the $N$ massless fields surrounding the black hole we use (61). Although we are not considering backreaction in this subsection, we can still evaluate $\mathcal{S}_P$ using the classical solution (48) to obtain the fine-grained entropy of the matter fields. In the Hartle-Hawking vacuum ($t_{\pm}(x^{\pm}) = 0$), the auxiliary field $\varphi$ is [see Eqs. (58) and (59)]

$$\varphi = -2\eta - \ln\left(-\lambda^2 x^+ x^-\right) + const. \qquad (70)$$

which means that there is a logarithmic divergence when evaluated at the horizon. However, its contribution to the entropy can be easily understood in terms of the entropy of a thermal bath. Indeed, if we rewrite the log term in light-cone coordinates

$$x^+ = \lambda^{-1}e^{\lambda\sigma^+}, \quad \text{and} \quad x^- = -\lambda^{-1}e^{-\lambda\sigma^-}, \qquad (71)$$

422   we obtain

$$-\frac{N}{12}\ln\left(-\lambda^2 x^+ x^-\right)\Big|_H = \frac{N}{12}\lambda(\sigma^- - \sigma^+)\Big|_H = \frac{N}{12}\lambda(2L) = \frac{N\pi}{6}T(2L) = \mathcal{S}_{thermal} \quad (72)$$

423   which is exactly the entropy of a thermal bath in an one dimensional box of length $2L = (\sigma^- - \sigma^+)|_H$,
424   as seen by an asymptotic observer. Since the length is infinite, $(\sigma^- - \sigma^+ = -2x \to \infty$ at the
425   horizon), the entropy diverges. This result allows us to rewrite the entropy of the Polyakov
426   term as

$$\mathcal{S}_P = \mathcal{S}_{quantum} + \mathcal{S}_{thermal}, \quad (73)$$

427   where

$$\mathcal{S}_{quantum} = \frac{N}{6}\eta|_H = -\frac{N}{12}\ln\left(\frac{M}{\lambda}\right). \quad (74)$$

428       The Polyakov entropy matches the fine-grained entropy given in eq. (93) of [37] for an eter-
429   nal black hole. This is easily done by identifying our $\mathcal{S}_{thermal}$ with the $\frac{N}{12}\ln\left(-x^+_{max}x^-_{max}/\delta^2\right)$
430   term. In ref. [37], the $x^{\pm}_{max}$ are infrared cut-offs for the right and left moving modes and
431   $\delta$ is a short distance cut-off introduced to regularize the logarithmic ultraviolet divergence
432   arising from the entanglement of the short-wavelength field fluctuations at the edge of the
433   horizon [54].
434       In what follows, we will subtract the thermal contribution to the entropy, since we want
435   to focus on vacuum effects. Now, the total entropy of the system is (excluding $\mathcal{S}_{thermal}$)

$$\mathcal{S}_{tot} = \mathcal{S}_{BH} + \mathcal{S}_{quantum} = \frac{2M}{\lambda} - \frac{N}{12}\ln\left(\frac{M}{\lambda}\right), \quad (75)$$

436   where we have used that, in the Kruskal gauge, $\phi = \eta$ is given by eq. (48).

437   ## 3.4   With backreaction, the BPP model

438   As we discussed, the backreaction on the metric of the quantum fluctuations of the matter
439   field can be considered by adding $I_P$ to the action $I_\phi$. However, the EoM from $I_\phi + I_P$ can not
440   be solved analytically. To address this problem, we can add the following extra term to the
441   action [40]

$$\begin{aligned} I_{extra} &= \frac{N}{24\pi}\int d^2x\sqrt{-g}\left[(1-2b)(\nabla\phi)^2 + (b-1)\phi R\right] \\ &= \frac{N}{24\pi}\int d^2x\left(-2(1-2b)\partial_+\phi\partial_-\phi + 4(b-1)\phi\partial_+\partial_-\eta\right). \end{aligned} \quad (76)$$

442   This results in a family of models characterized by the parameter $b$. $I_{extra}$ being local modifies
443   the local dynamics but not the global properties. For $b = 1/2$ we recover the RST model [38]
444   and for $b = 0$ we recover the BPP model [39]. By introducing the Liouville fields [40]

$$\Omega = \sqrt{\frac{N}{12}}b\phi + \sqrt{\frac{12}{N}}e^{-2\phi}, \quad (77)$$

$$\chi = \sqrt{\frac{N}{12}}\eta + \sqrt{\frac{N}{12}}(b-1)\phi + \sqrt{\frac{12}{N}}e^{-2\phi}, \quad (78)$$

445   we can rewrite our family of models as a Liouville theory that can be solved analytically. The
446   EoM for the action $I_\phi + I_P + I_{extra}$ in terms of the Liouville variables are

$$\partial_+\partial_-\chi = -\lambda^2\sqrt{\frac{12}{N}}e^{\sqrt{\frac{48}{N}}(\chi-\Omega)}, \quad (79)$$

$$\partial_+\partial_-\Omega = -\lambda^2\sqrt{\frac{12}{N}}e^{\sqrt{\frac{48}{N}}(\chi-\Omega)}, \quad (80)$$

which implies

$$\partial_+\partial_-(\chi - \Omega) = 0. \tag{81}$$

The constraint equations become

$$-\partial_\pm\chi\partial_\pm\chi + \sqrt{\frac{N}{12}}\partial_\pm^2\chi + \partial_\pm\Omega\partial_\pm\Omega - \frac{N}{12}t_\pm = 0. \tag{82}$$

In the Kruskal gauge $\Omega = \chi$ and for the Hartle-Hawking vacuum ($t_\pm = 0$) we have the solution

$$\sqrt{\frac{N}{12}}\Omega = \frac{M}{\lambda} - \lambda^2 x^+ x^-. \tag{83}$$

From now on, we focus on the model $b = 0$, i.e. the BPP model, since it results in a simpler solution. Taking $b = 0$ and (83) into the Liouville variables (77) and (78), it is immediate to find

$$\phi = \eta = -\frac{1}{2}\ln\left(\frac{M}{\lambda} - \lambda^2 x^+ x^-\right). \tag{84}$$

These solutions are the same as the classical solutions (48), which indicates that the metric and the dilaton do not have quantum corrections in the BPP model.

### 3.4.1 Null energy condition

The results for the NEC in section 3.3.1 don't change when we consider the BPP model in the Hartle-Hawking vacuum since the metric is not affected by the quantum corrections. Figure 4 summarizes the results for the NEC in the BPP model and the classical results.

### 3.4.2 Entropy and thermodynamics

The evaluation of the entropy follows the discussion in section 3.3.2. Since we included the $I_{bpp} = I_{extra}(b = 0)$ term in the action, we have an additional contribution to the entropy

$$\mathcal{S}_{bpp} = \frac{4\pi}{\sqrt{-g}}\frac{\partial\mathcal{L}_{bpp}}{\partial R}\bigg|_H = -\frac{N}{6}\phi\bigg|_H = \frac{N}{12}\ln\left(\frac{M}{\lambda}\right). \tag{85}$$

The total entropy of the semi-classical system is (as before, we omit the $\mathcal{S}_{thermal}$ contribution of $\mathcal{S}_P$)

$$\mathcal{S}_{tot} = \mathcal{S}_\phi + \mathcal{S}_{bpp} + \mathcal{S}_{quantum} = 2e^{-2\phi}\bigg|_H = \frac{2M}{\lambda}, \tag{86}$$

where we have used that, in the Kruskal gauge $\phi = \eta$, and the dilaton is given by (48). Note that $\mathcal{S}_{bpp}$ cancels with $\mathcal{S}_{quantum}$. We find that $\mathcal{S}_{tot}$ is equal to the $\mathcal{S}_{BH}$ for the case without backreaction (69). It means that the total entropy of the semi-classical system is exactly the entropy of a dilatonic black hole of mass $M$ at a temperature $\lambda/2\pi$. This is not surprising since the solution to the semi-classical equations in the Hartle-Hawking vacuum state is precisely a black hole of mass $M$ at a temperature $\lambda/2\pi$. However, we can still split the total entropy into two parts: the entropy of the black hole, and the entropy of the matter sector

$$\mathcal{S}_{BH} = \mathcal{S}_\phi + \mathcal{S}_{bpp} = \frac{2M}{\lambda} + \frac{N}{12}\ln\left(\frac{M}{\lambda}\right), \tag{87}$$

$$\mathcal{S}_{quantum} = -\frac{N}{12}\ln\frac{M}{\lambda}. \tag{88}$$

The matter entropy $\mathcal{S}_{quantum}$ is negative just because we removed the log divergent contribution coming from the thermal bath. Let us note that $\mathcal{S}_{BH}$ has a quantum correction as compared to the black hole entropy without backreaction (69) coming from $\mathcal{S}_{bpp}$.

474  We finally comment on a related quantity, which is the black hole heat capacitance for the
475 complete system (see Eqs. (86) and (72))

$$C_{tot} = T \frac{\partial (\mathcal{S}_{tot} + \mathcal{S}_{thermal})}{\partial T} = \frac{\pi N T L}{3} - \frac{M}{\pi T} \,, \tag{89}$$

476 and which is consistently positive, due to the dominant contribution of the thermal bath. Thus
477 the full system is thermodynamically stable.

## 4 Comparison

479 In sections 2 and 3 we discussed the thermodynamics of finite volume effects in QFT and dila-
480 tonic black holes respectively. These effects seem at first glance unrelated. However, in both
481 systems we have NEC violation and finite quantum entropy. But the analogy runs even deeper;
482 we observe that the effect of the temperature $T$ in black holes is similar to the effect of the
483 inverse length scale $1/a$ in tunneling. To see this, we compare two physical quantities, which
484 are the NEC and the rate of change in entropy, when the environment is modified.
485
486 *NEC violation.*
487 For tunneling, the NEC was computed in section 2.3. We will compare in the regime where
488 $ma \lesssim 1$, where the NEC becomes

$$NEC_{\text{tun}} \approx -\frac{\pi}{3a^2} \,. \tag{90}$$

489 It is interesting to note that in this regime the Casimir effect is dominant over the tunneling
490 effects. The result on the black hole horizon for any ratio $T/M$ in the BPP model is computed
491 in section 3

$$NEC_{\text{BH}} = -\frac{N \pi^2 T^2}{12} \,. \tag{91}$$

492 We notice the clear equivalence of the dominant terms in the regime of interest: NEC violation
493 is proportional to $1/a^2$ and $T^2$ for the tunneling and the black hole respectively. Physically,
494 this is partly a manifestation of the effect that the NEC violation is due to finite volume effects
495 in one case and the non-zero temperature of the black hole in the other.
496
497 *Entropy*
498 For the tunneling case we found (Sec. 2.4)

$$\mathcal{S}_{tun} = \ln(m\beta) - \frac{2ma}{3} + \frac{1}{2} \ln \left( \frac{4ma}{\pi} \right) \,. \tag{92}$$

499 The black hole entropy is (Sec. 3.4.2)

$$\mathcal{S}_{\text{BH}} + \mathcal{S}_{thermal} = \frac{N\pi}{6} T(2L) + \frac{M}{\pi T} + \frac{N}{12} \ln \left( \frac{M}{2\pi T} \right) \,. \tag{93}$$

500 We note that both entropies have a term that diverges in the limit of zero temperature for
501 the tunneling and the limit of infinite volume for the black hole. These terms are $\ln(m\beta)$ for
502 tunneling and $\frac{N\pi}{6} T(2L)$ for the black hole. We will remove these terms as we want to compare
503 the quantum entropies of the two systems without infinite contributions. Then the entropy in
504 both cases can be negative.
505   Instead of comparing the entropies as they are, we will focus on the rate of change of the
506 entropy in terms of the relevant parameter $w$ for each system

$$R \equiv w \frac{\partial \mathcal{S}}{\partial w} \,. \tag{94}$$

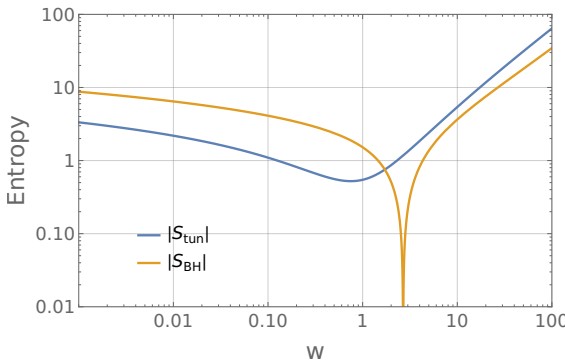

Figure 5: Tunneling (92) and black hole (93) entropies as a function of w = $ma$ and w = $\frac{M}{T}$, respectively. We have removed the $\ln(m\beta)$ divergence and set $N = 12$.

So we consider the rate of change in entropy when the length $a$ varies

$$R_{\text{tun}} \equiv a \frac{\partial \mathcal{S}_{\text{tun}}}{\partial a} = \frac{1}{2} - \frac{2m}{3}a \ . \tag{95}$$

For the black hole case, the rate of change is in terms of the temperature

$$R_{BH} \equiv T \frac{\partial \mathcal{S}_{\text{BH}}}{\partial T} = -\frac{N}{12} - \frac{M}{\pi T} \ . \tag{96}$$

As for NEC violation, we observe that there is a clear equivalence of the rates of change in entropy when one maps $a$ to $1/T$. This is not an effect that can be explained by simple dimensional analysis: the quantities $ma$ and $T/M$ are dimensionless and thus any combination could appear in these expressions.

Fig. 5 shows the absolute value of $\mathcal{S}_{tun}$ and $\mathcal{S}_{BH}$ as a function of $ma$ and $M/T$, respectively. Without the term $\ln(m\beta)$, $\mathcal{S}_{tun}$ is always negative because of the dominant contribution of the negative linear term. For $\mathcal{S}_{BH}$, the positive linear term becomes dominant when $M/T$ exceeds $\frac{N\pi}{12}W_0(\frac{24}{N})$. The divergence observed in the logarithmic plot indicates a change in the sign of $\mathcal{S}_{BH}$ at higher temperatures, where the logarithmic term takes over. Although these quantities can be negative, the total entropy remains positive due to the contributions of $\ln(m\beta)$ and $S_{thermal}$.

We should briefly comment on a relevant comparison between the Casimir effect and black hole thermodynamics in refs. [57, 58]. There, the authors used isothermal compressibility instead of the rate of change of the entropy but also found an analogy.

## 5   Conclusions

Motivated by the description of Hawking radiation in terms of NEC violation, we mapped a few aspects of black hole thermodynamics with finite volume effects in QFT arising from tunnelling in a confined space. Both descriptions are done in $1 + 1$ dimensions, and feature similar behaviours in terms of energy and entropy, when one identifies the inverse of the BH temperature with the finite size of the confining space for a scalar field. The origin of the mapping is the presence of boundaries in both systems, either in the form of a horizon for the BH, or in the form of periodic boundary conditions in QFT.

We focused the comparison on the regime $ma \propto M/T \lesssim 1$, and not on the regime $ma \propto M/T \gg 1$. In the latter case, the non-trivial effects vanish exponentially with $a$ for tunnelling, whereas for the black hole they vanish as a power law with $T$. In this example, the

mapping we discussed does not hold and a more thorough discussion is necessary to include all the regimes in this study, which is left for a future work.

One possibility is the approach described in [59], where a massless scalar field on a $D$-sphere is considered, and a non-minimal coupling to curvature is introduced, which provides an effective mass. The latter depends on the curvature, such that the action of the instanton is not proportional to the volume, and therefore tunnelling is not suppressed exponentially with the volume. In particular, for $D = 3$ NEC violation varies as $1/a$ for all values of $a$, and such a power law dependence is more likely to match the black hole description. This approach does not modify the picture presented here, but further studies involving $D \geq 2$ space dimensions are necessary.

An essential point in our study is the presence of an environment which is necessary to justify the static regimes we study. For tunnelling, this environment fixes the spatial period, whereas for the black hole it plays the role of a heat source and fixes the temperature. Removing this environment could also be an avenue to explore, in which case the equilibrium assumption is no longer valid; since the black hole evaporates and the confining space for the scalar field is modified by energetics of NEC violation.

Another interesting connection point that was not discussed in this work, is that Hawking radiation can be studied as tunneling of particles through the black hole horizon [60]. Then both systems can be viewed as tunneling, one on a flat background with a finite volume and one on a curved background with an infinite volume. This connection could be explored further in future work, especially in the $3 + 1$ dimensional case.

Given the above analogies, we hope to find a more formal description for this mapping in light of the AdS/CFT correspondence, although both systems here have the same dimensionality. Hopefully such a mapping could be extended to 3+1 dimensions, and might be relevant to either astrophysics or analogue condensed matter systems.

# Acknowledgements

The authors would like to thank Dionysios Anninos, Jose Navarro-Salas, and Andrew Svesko for useful discussions.

**Funding information** The Science and Technology Facilities Council supports the work of JA and DB (grant No. STFC-ST/X000753/1), the Engineering and Physical Sciences Research Council supports the work of JA (grant No. EP/V002821/1) and the Leverhulme Trust supports the work of JA and SP (grant No. RPG-2021-299). DPS is supported by the EPSRC studentship grant EP/W524475/1. For the purpose of Open Access, the authors have applied a CC BY public copyright licence to any Author Accepted Manuscript version arising from this submission.

# A   Casimir effect for static saddle points

This appendix is based on the book [9], and we consider periodic boundary conditions in space. We first show the derivation for vanishing temperature (limit $\beta \to \infty$), for which the ground state energy contains an ultraviolet divergence. We then show how to include finite-temperature effects, which do not introduce new divergences.

Evaluating the individual connected graph generating functional $W[\phi_i]$ (10) at the static

saddle points $\phi_s = \pm 1$ for vanishing source $j = 0$ yields

$$W[\phi_s] \equiv W_{\text{stat}}(a, \beta) = \frac{1}{2} \sum_{n \in \mathbb{Z}} \sum_{l \in \mathbb{Z}} \ln\left( \frac{\nu_l^2 + \omega_n^2}{\nu_l^2} \right) , \tag{A.1}$$

where $\omega_n = \sqrt{m^2 + k_n^2}$, and

$$\nu_l = \frac{2\pi l}{\beta} \quad , \quad k_n = \frac{2\pi n}{a} . \tag{A.2}$$

The origin of energies is chosen in such a way as to recover the usual sum of ground state energies of harmonic oscillators, at zero temperature.

## A.1   Zero temperature

In the limit of zero temperature the summation over Matsubara modes becomes an integral

$$\lim_{\beta \to \infty} \{W[\phi_s]\} = \frac{\beta}{2} \sum_{n \in \mathbb{Z}} \int_{-\infty}^{\infty} \frac{d\nu}{2\pi} \ln\left( \frac{\nu^2 + \omega_n^2}{\nu^2} \right) = \frac{\beta}{2} \sum_{n \in \mathbb{Z}} \omega_n . \tag{A.3}$$

Using the Abel-Plana formula

$$\sum_{n \in \mathbb{N}} F(n) \equiv -\frac{1}{2} F(0) + \int_0^{\infty} F(t) dt + i \int_0^{\infty} \frac{dt}{e^{2\pi t} - 1} \left[ F(it) - F(-it) \right] , \tag{A.4}$$

the sum over the frequencies can be expressed as

$$\sum_{n \in \mathbb{Z}} \omega(n) = 2a\Lambda^2 + 2i \int_0^{\infty} \frac{dt}{e^{2\pi t} - 1} \left[ \omega(it) - \omega(-it) \right] , \tag{A.5}$$

where the ultraviolet divergence is

$$\Lambda^2 \equiv \frac{m^2}{2\pi} \int_0^{\infty} \sqrt{t^2 + 1} \, dt . \tag{A.6}$$

By considering the principle branch $z \in ]-\infty, 0]$ of $\ln(z)$, we have

$$\omega(it) - \omega(-it) = \frac{4\pi i}{a} \sqrt{t^2 - \mu^2} \theta(t - \mu) , \tag{A.7}$$

where $\mu \equiv ma/2\pi$. The individual connected graph generating functional for static saddle points therefore reads, in the limit of zero temperature,

$$\lim_{\beta \to \infty} \{W[\phi_s]\} = a\beta\Lambda^2 - \beta \frac{4\pi}{a} \int_{\mu}^{\infty} \frac{dt}{e^{2\pi t} - 1} \sqrt{t^2 - \mu^2} \tag{A.8}$$

$$= \beta a \Lambda^2 - \beta \frac{m^2 a}{\pi} \int_1^{\infty} \frac{du}{e^{mau} - 1} \sqrt{u^2 - 1} . \tag{A.9}$$

## A.2   Finite-temperature corrections

In order to calculate thermal corrections to the individual connected graph generating functional for static saddle points, we employ the zeta-function regularisation and write the expression (11) as

$$W_{\text{stat}}(a, \beta, s) = -\frac{1}{2} \frac{\partial}{\partial s} \left( \sum_{l \in \mathbb{Z}} \sum_{n \in \mathbb{Z}} (\beta a)^{-s} \left( \nu_l^2 + \omega_n^2 \right)^{-s} \right) , \tag{A.10}$$

where the ultraviolet divergence is turned into a divergence in the limit $s \to 0$. The above can be written in terms of a parametric integral

$$W_{\text{stat}}(a, \beta, s) = -\frac{1}{2} \frac{\partial}{\partial s} \left( \int_0^\infty \frac{dt}{t} \frac{t^s}{\Gamma(s)} \sum_{l \in \mathbb{Z}} \sum_{n \in \mathbb{Z}} e^{-t\beta a \left(\nu_l^2 + \omega_n^2\right)} \right) . \tag{A.11}$$

From the Poisson summation formula, one can derive the following identity

$$\sum_{l \in \mathbb{Z}} e^{-zl^2} = \sqrt{\frac{\pi}{z}} \sum_{l \in \mathbb{Z}} e^{-\pi^2 l^2 / z} , \tag{A.12}$$

which, when applied to the Matsubara sum in eq. (A.11) with $z = \beta a t (2\pi T)^2$, leads to

$$W_{\text{stat}}(a, \beta, s) = -\frac{\beta}{2} \frac{\partial}{\partial s} \left( \sum_{l \in \mathbb{Z}} \int_0^\infty \frac{dt}{t} \frac{t^s}{\Gamma(s)\sqrt{4\pi\beta a t}} \sum_{n \in \mathbb{Z}} e^{-\frac{l^2 \beta^2}{4\beta a t} - \beta a t \omega_n^2} \right) . \tag{A.13}$$

The ultraviolet divergence is contained within the temperature-independent integral for $l = 0$. We thus make the following decomposition of eq. (A.13)

$$W_{\text{stat}}(a, \beta, s) = \lim_{\beta \to \infty} \left\{ W_{\text{stat}}(a, \beta, s) \right\} + W_{\text{stat}}^T(a, \beta) , \tag{A.14}$$

where the temperature independent part is calculated in the previous section, and the temperature-dependent part is given by

$$W_{\text{stat}}^T(a, \beta) \equiv -\frac{\beta}{\sqrt{4\pi\beta a}} \sum_{l \in \mathbb{N}} \int_0^\infty \frac{dt}{t^{3/2}} \sum_{n \in \mathbb{Z}} e^{-\frac{l^2 \beta^2}{4\beta a t} - \beta a t \omega_n^2} . \tag{A.15}$$

Note that the regulator has been removed from eq. (A.15) using

$$\lim_{s \to 0} \left[ \frac{\partial}{\partial s} \frac{f(s)}{\Gamma(s)} \right] = f(0) . \tag{A.16}$$

The integral and summation over $l$ in eq. (A.15) can be then evaluated, leading to

$$W_{\text{stat}}^T(a, \beta) = \sum_{n \in \mathbb{Z}} \ln \left( 1 - e^{-\beta \omega_n} \right) . \tag{A.17}$$

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
