# Peer review of "Mapping 1+1-dimensional black hole thermodynamics to finite volume effects"

_SciPost Physics_

## Round 1 · Referee Report · Anonymous (Referee 1) · 2025-2-14

Report

In this paper, the authors analyse two quantum systems, one being a massive particle on a circle, and the other - a 2D black hole. They compute the NEC violation and the entropy rate and both system and find that they behave similarly in certain regimes and with certain identification of the parameters. This observation is the main result of the paper.

I do not have any objections to the treatment of these particular quantum systems. However, the conclusion made by the authors does not sound conclusive to me. There are two reasons. One, of a general nature, is that perhaps in any two systems one can find two quantities of similar behaviour after looking sufficiently hard for suitable regimes and suitable identifications of the parameters. The second one is that in the 2D gravity model considered by the authors the black temperature is independent of the mass, see eq. (49). The corresponding thermodynamics looks too exotic to draw general conclusions.

In my opinion, the authors should present some good arguments that the effect that they observed is a universal one independent of the choice of particular models.

Recommendation

Ask for major revision

---

## Editorial Decision

awaiting_resubmission